# Synthesis of Galectin Inhibitors by Regioselective 3′-*O*-Sulfation of Vanillin Lactosides Obtained under Phase Transfer Catalysis

**DOI:** 10.3390/molecules26010115

**Published:** 2020-12-29

**Authors:** Karima Belkhadem, Yihong Cao, René Roy

**Affiliations:** 1Department of Chemistry, University of Québec à Montréal, P.O. Box 8888, Succ. Centre-Ville, Montréal, QC H3C 3P8, Canada; karimachimi@hotmail.com (K.B.); yihongcao1986@gmail.com (Y.C.); 2INRS-Institut Armand-Frappier, Université du Québec, 531 boul. des Prairies, Laval, QC H7V 1B7, Canada

**Keywords:** lactosides, galectins, vanillin, cancer, sulfation, phase transfer catalysis, Knoevenagel-Doebner reaction

## Abstract

Vanillin-based lactoside derivatives were synthetized using phase-transfer catalyzed reactions from per-*O*-acetylated lactosyl bromide. The aldehyde group of the vanillin moiety was then modified to generate a series of related analogs having variable functionalities in the *para-* position of the aromatic residue. The corresponding unprotected lactosides, obtained by Zemplén transesterification, were regioselectively 3′-*O*-sulfated using tin chemistry activation followed by treatment with sulfur trioxide-trimethylamine complex (Men_3_N-SO_3_). Additional derivatives were also prepared from the vanillin’s aldehyde using a Knoevenagel reaction to provide extended α, β-unsaturated carboxylic acid which was next reduced to the saturated counterpart.

## 1. Introduction

Cellular communications are frequently governed by molecular interactions involving cell surface glycoconjugates overlay expressed as glycosaminoglycans (GAGs), glycoproteins, and glycolipids. Of particular interest are the family of S-type lectins represented by galectins (Gal) that are characterized by their carbohydrate recognition domains (CRDs) having affinity for glycoconjugates with exposed β-d-galactopyranoside residues in common [1,2,3]. So far, 15 family members have been identified in mammals. In spite of their similar characteristic sugar binding recognition patterns, they are distinctly divided into three categories depending on their molecular architectures. They are classified as: a) Prototype dimers (Gal-1, -2, -5, -7, -10, -11, -13, -14, -15); b) tandem repeat (Gal-4, -6, -8, -9, -12); and c) monomeric chimera type capable of oligomerization (Gal-3) [4]. Given that they are expressed intracellularly together with being present extracellularly by secretion and that they are critically implicated in a plethora of physiological functions, including cancer, inflammation, and immune responses, discovery of specific inhibitors has become of keen therapeutic interest, albeit a major challenge in medicinal chemistry [4,5,6,7,8,9].

Consequently, the design of highly selective sugar-based inhibitors against each of the galectins has been the subject of intense research activities. The field is however only dominated by a few research groups [4,5,6,7,8,9] amongst which the team of Leffler/Nilsson being clearly dominating with a number of successful glycomimetics, some of which reaching clinical phases (thiodigalactosides, TD139, GB1107) [10,11,12,13]. Essentially based on β-d-galactopyranoside/lactoside/*N*-acetyllactosamine lead scaffolds [14], the incorporation of pharmacophores have provided the most successful candidates when appended at either the anomeric position [9,15,16,17,18,19,20,21,22] or on position -3 of the galactopyranosides [10,11,12,13] residue or *O*-3′ in the case of lactosides [4,23]. Appealingly, introduction of a negative sulfate group at O-3/O-3′ has also afforded potent ligands owing to the presence of charged amino acids within the CRD [15,23,24,25,26,27]. In addition to the above chemical modifications, an additional and quite successful synthetic strategy has been the discovery that multivalent galactosides/lactosides in the form of glycodendrimers [28,29,30,31,32,33,34], glycopolymers [35,36], liposomes/dendrimersomes [37,38,39,40], and protein conjugate with appended TD139 [41] could similarly provide gains in both affinity and selectivity. This was predominantly observed for the chimeric galectin-3 that can oligomerize upon binding to multivalent receptors due to its collagenous peptide tail [29].

In this paper, we aimed to combine both modifications to a lactopyranoside scaffold by incorporating aromatic aglycons simultaneously to an O-3′ sulfate group (Figure 1). This combined choice was dictated by the fact that a few galectin members have been shown to bind preferentially to sulfated glycans. This was particularly through for Gal-1 [23,24], -3 [15], -4 [25,26], and -8 [27]. In the latter case, recent modeling and X-ray experiments demonstrated that the beneficial interactions were due to favorable electrostatic interactions with arginine residues (Arg-45 and Arg-59) [4,27]. Furthermore, even though several hydrophobic aglycons have been advantageous in the binding events using affinity measurements by ITC, we choose the natural vanillin as aglycon because of its well-established lack of toxicity, the presence of a large amount of vanillin glucoside (glucovanillin) in foodstuffs [42], and the well-known antioxidant properties of phenolic glycosides.

## 2. Results and Discussion

Over the years, Phase Transfer Catalysis (PTC) has emerged as an efficient and practical methodology for stereoselective glycosidation [43,44] and related anomeric substitutions. Under our previously optimized conditions for phenolic glycosides (TBAHS, EtOAc, 1M Na_2_CO_3_), we treated peracetylated lactosyl bromide **1** [43] with vanillin derivatives **2**, **3**, and **4** to afford lactosides **5**, **7**, and **8** in 80, 66, and 58% yields, respectively (Scheme 1). The major by-product of the reactions was the usual 2-acetoxy-lactal (^1^H-NMR of H1 at δ 6.61 ppm) resulting from the HBr S_N_2 elimination process [43,44]. The disappearance of the anomeric doublet of the bromide **1** at δ 6.52 ppm with a J_1,2_ coupling constant of 4.0 Hz was replaced by a new doublet at δ 5.10 ppm with a distinctive *trans* coupling constant of J_1,2_ of 7.4 Hz in compound **5**. The same held for methyl ester **7** (H1: δ 5.05, J_1,2_ = 7.4 Hz) and *tert*-butyl ester **8** (H1: δ 5.03, J_1,2_ = 7.5 Hz). As previously demonstrated [42,43,44], these PTC conditions afforded complete stereoselectivity in favor of anomeric inversion from the α-bromide **1** to β-glycosides **5**, **7**, and **8** exclusively. Since we also wanted to explore the role played by the carboxylic function in the *para*-position of the vanillin residue, coupled to the fact that we could not hydrolyze the acetate protecting groups in derivative **7** and **8** without losing the esters (*vide infra*), we opted for the oxidation of the native vanillin’s aldehyde in **5** using permanganate treatment (KMnO_4_, H_2_O, 80 °C, 20 min) which afforded acid **6** in 52% yield, showing disappearance of the aldehyde proton at δ 9.91 ppm. All compounds were fully characterized by NMR spectroscopy (^1^H, ^13^C), mass spectrometry, and the datasets agreed with literature data when known (see experimental section).

The peracetylated intermediates **5**, **7**, **8** were next de-*O*-acetylated under the classical Zemplén conditions (NaOMe, MeOH) in essentially quantitative yields in all cases to give free lactosides **9**, **13**, and **15** (Scheme 2). Following sequential treatment of the unprotected lactosides with dibutyltin oxide (Bu_2_SnO, DMF, PhMe, 90° C, 6h) and regioselective sulfation [45] using sulfur trioxide-trimethylamine complex (Me_3_N^.^SO_3_), the 3′-*O*-sulfated lactosides **10**, **14**, **16** (Scheme 2) were obtained in good to excellent yields (81%-quantitative). The regioselectivity of this transformation is well-known and has been explained through the formation of a cyclic stannylene complex at the unique *cis*-3′,4′-dihydroxyl groups of the galactoside moiety [46,47]. The aldehyde functions of vanillin lactoside **9** and 3′-*O*-sulfated lactoside **10** were reduced using NaBH_4_ in MeOH (rt, 3h) to give vanillin lactoside analogs **11** and **12** in excellent 95% and 88% yields, respectively. The position of the sulfate groups were readily confirmed on the basis of the H-3′ downfield shift from ~3.66 ppm to ~4.28 ppm (SI) together with the characteristic ^13^C-NMR chemical shift displacement of the 3′-carbon, which usually appears ~7 ppm downfield (~ δ 80 ppm) from the unsubstituted precursors at ~δ 73 ppm [47]. Dept 135 ^13^C-NMR analysis was required to ensure that the regioselective sulfation was unequivocally performed at the C-3’ position in order to show the absence of signals attributed to C-6 and C-6’s modifications (usually at δ 59–62 ppm [47]. NMR COSY and HSQC experiments were also used to unambiguously correlate the sulfation process and regioselectivity.

A Knoevenagel-Doebner condensation was also used (malonic acid, C_5_H_5_N, Piperidine, 95° C, 3h) for the synthesis of two carbons homologated vanillin lactosides **17**–**20** (Scheme 3), analogously used before for a galactoside derivative [48]. Thus, aldehyde **5** provided peracetylated α, β-unsaturated analog **17** in 65% yield, which upon catalytic hydrogenation (H_2_, Pd-C, THF-MeOH, rt, 5 h) gave the expected α, β-saturated lactoside **19** essentially quantitatively.

The ^1^H-NMR spectra of **17** clearly showed the two *trans* proton signals corresponding to the unsaturated protons H-14 and H-13 at δ 6.33 and 7.68 ppm with a typical coupling constant J_13-14_ of 15.9 Hz. The palladium-catalyzed reduction leading to **19** showed the disappearance of these two proton signals and the appearance of new H-14 and H-13 protons at 2.62 and 2.89 ppm, respectively (see Appendix A). Zemplén deprotection was used to afford the de-*O*-acetylated lactosides **18** and **20** (Scheme 3).

## 3. Materials and Methods

### 3.1. General Synthetic Methods

All reactions in organic medium were performed in standard oven dried glassware under an inert atmosphere of nitrogen using freshly distilled solvents. Solvents and reagents were deoxygenated, when necessary by purging with nitrogen. All reagents were used as supplied without prior purification unless otherwise stated, and obtained from Sigma-Aldrich Chemical Co. Ltd.(St. Louis, MO, USA) Reactions were monitored by analytical thin-layer chromatography (TLC) using silica gel 60 F254 precoated plates (E. Merck (Darmstadt, Germany)) and compounds were visualized with a 254 nm UV lamp, a mixture of iodine/silica gel and/or mixture of ceric ammonium molybdate solution (100 mL H_2_SO_4_, 900 mL H_2_O, 25 g (NH_4_)_6_Mo_7_O_24_H_2_O, 10 g Ce(SO_4_)_2_), and subsequent spots development by gentle warming with a heat-gun. Purifications were performed by silica gel flash column chromatography using Silica (60 Å, 40–63 µm) with the indicated eluent. NMR spectroscopy was used to record ^1^H-NMR and ^13^C-NMR spectra at 300 or 600 MHz and at 75 or 150 MHz, respectively, on Bruker (300 MHz) and Bruker Avance III HD 600 MHz spectrometers (Billerica, MA, USA). Proton and carbon chemical shifts (δ) are reported in ppm relative to the chemical shift of residual CHCl_3_, which was set at 7.26 ppm (^1^H) and 77.16 ppm (^13^C). Coupling constants (*J*) are reported in Hertz (Hz), and the following abbreviations are used for peak multiplicities: Singlet (s), doublet (d), doublet of doublets (dd), doublet of doublet with equal coupling constants (t_ap_), triplet (t), multiplet (m). Analysis and assignments were made using COSY (Correlated SpectroscopY) and HSQC (Heteronuclear Single Quantum Coherence) experiments. High-resolution mass spectrometry (HRMS) data were measured with a LC-MS-TOF (Liquid Chromatography-Mass Spectrometry-Time of Flight; Agilent Technologies) in positive and/or negative electrospray mode(s) at the analytical platform of UQAM.

### 3.2. General Synthetic Procedure A: Phase-Transfer Catalysis (PTC) Reaction

PTC reactions were performed following the previously established protocols [42,43,44,45,46] or under the slightly modified procedure as follows: To a solution of peracetylated lactosyl bromide **1** [49] (1 equiv.) in ethyl acetate (6 mL) was added the corresponding aromatic alcohol (2.5 equiv.), tetrabutylammonium hydrogen sulfate (TBAHS, 1.1 equiv.) and 1M Na_2_CO_3_ (1.3 equiv.). The mixture was stirred at room temperature for 2h30 min and then washed successively with water and brine. The organic layer was dried over Na_2_SO_4_ and concentrated under reduced pressure. Purification by silica gel column chromatography (Hex/AcOEt) afforded the corresponding compounds **5, 7,** and **8** as white powders (yield 58–80%).

### 3.3. General Synthetic Procedure B: Zemplén Transesterification Reaction

To a solution of lactosides **5, 6, 7, 8**, **17** and **19** in dry methanol was added a solution of sodium methoxide (1 M in MeOH, 0.1 equiv.). After stirring at room temperature for 1–2 h, the reaction was completed and then neutralized by addition of ion-exchange resin (Amberlite IR 120 H^+^). The solution was filtered and evaporated in vacuo to afford the de-*O*-acetylated lactosides as white powders (yield 95%-quant.)

### 3.4. General Synthetic Procedure C: Preparation of 3′-*O*-sulfated Lactosides

A mixture of deacetylated lactosides (1 equiv.) and dibutyltin oxide (Bu_2_SnO, 1.15 equiv.) in DMF/toluene (6 mL/3 mL) was stirred at 90 °C for 6 h. The solution was then concentrated and sulfur trioxide-trimethylamine complex (Me_3_N^.^SO_3_) (1.3 equiv.) and dry DMF (6 mL) were added. After stirring at room temperature for 17 h, the reaction was quenched with water and evaporated under vacuum. The residue was purified through a column of DOWEX Marathon C (Na^+^) and eluted with H_2_O to obtain the pure 3′-*O*-sulfated lactosides as white powder after lyophilization (yields 82%-quantitative).

### 3.5. 2,3,4,6-Tetra-*O*-acetyl-β-d-galactopyranosyl-(1-4)-2,3,6-tri-*O*-acetyl-α-d-glucopyranosyl bromide (**1**) (Acetobromolactose)

To a solution of per-*O*-acetylated lactose [49] (14.2 g, 21 mmoL) in anhydrous CH_2_Cl_2_ (63 mL) was added hydrobromic acid (33% in AcOH, 47.9 mL). The reaction mixture was stirred at room temperature for 1 h, then neutralized with saturated aqueous NaHCO_3_ and washed with brine. The organic layer was dried over Na_2_SO_4_ and concentrated under reduced pressure to give lactosyl bromide **1** (13.6 g, 93%) as a white solid. Its spectroscopic data agreed well with those of the literature [49].

### 3.6. 3-Methoxy-4-(2,3,6,2′,3′,4′,6′-hepta-*O*-acetyl-β-D-lactopyranosyloxy)benzaldehyde (**5**) (4-formyl-2-methoxyphenyl 2,3,6,2′,3′,4′,6′-hepta-*O*-acetyl-β-D-lactopyranoside)

Following the general procedure A, compound **5** was obtained as a white powder; yield: 176 mg (80%). ^1^H-NMR (300 MHz, CDCl_3_): δ (ppm) 9.91 (s, 1H, CHO), 7.38–7.46 (m, 2H, H_arom_), 7.19 (d, 1H, *J* = 8.0 Hz,H_arom_), 5.38 (d,1H, *J* = 2.8 Hz, H-4′), 5.28 (dd, 2H, *J* = 13.9,*J* = 8.0 Hz, H-3 et H-2), 5.17(dd, 1H, *J* = 14.7, *J* = 6.9 Hz, H-2′), 5.10 (d, 1H, *J* = 7.4 Hz, H-1), 4.99 (dd,1H,*J* = 10.4, *J* = 3.4 Hz, H-3′), 4.52–4.57 (m, 2H, H-1′, H-6a), 4.04–4.24(m, 3H, H-6b, H-6′ab), 3.91–3.96 (m, 2H, H-4 et H-5′), 3.90 (s, 3H, OMe), 3.76–3.81 (m, 1H, H-5), 1.94–2.22 (21H, 7OAc). ^13^C-NMR (75 MHz, CDCl_3_): (75 MHz, CDCl_3_): δ (ppm) 190.9 (CHO), 170.3, 170.2, 170.1, 170.0, 169.7, 169.5, 169.0 (CO), 151.1, 150.8, 132.6, 125.3, 117.7, 110.6 (C_arom_), 101.1 (C-1′), 99.3 (C-1), 76.0 (C-4), 72.9 (C-5), 72.4 (C-3), 71.2 (C-2), 70.9 (C-3′), 70.7 (C-5′), 69.0 (C-2′), 66.5 (C-4′), 61.8 (C-6′), 60.8 (C-6), 56.0 (OMe), 20.8, 20.7, 20.6, 20.5.

### 3.7. 3-Methoxy-4-(2,3,6,2′,3′,4′,6′-hepta-*O*-acetyl-β-d-lactopyranosyloxy)benzoic Acid (**6**)

A solution of compound **5** (50 mg, 0.063 mmol) in 1 mL of water and in the presence of KMnO4 (12.04 mg, 0.076 mmol), which was added drip under agitation and refluxed of 70–80 °C for about 20 min. The residue is washed with hot water, filtered, concentrated, and then acidified to give the compound **6** was obtained as a white powder; yield: 26.5mg (52%), ^1^H-NMR (300 MHz, CDCl_3_): δ (ppm) 7.67 (d, 1H, *J* = 8.4 Hz, H_arom_), 7.61 (s, 1H, H_arom_),7.19 (d, 1H, *J* = 8.0 Hz, H_arom_),5.36 (d,1H, *J* = 2.8 Hz, H-4′), 5.26(ddd,2H, *J* = 13.9,*J* = 8.0 Hz, H-3 et H-2), 5.12(dd, 1H, *J* = 10.3, *J* = 7.9 Hz, H-2′), 5.05 (d, 1H, *J* = 7.3 Hz, H-1), 4.97 (dd,1H,*J* = 10.4, *J* = 3.4 Hz, H-3′), 4.50–4.54 (m, 2H, H-1′,H-6a), 4.03–4.21 (m, 3H, H-6b, H-6′ab), 3.89–3.91 (m, 2H, H-4 et H-5′), 3.87 (s, 3H, OMe), 3.69–3.81 (m, 1H, H-5), 1.81–2.34 (21H, 7OAc). ^13^C-NMR (75 MHz, CDCl_3_): (75 MHz, CDCl_3_): 170.4, 170.3, 170.1, 170.1, 169.8, 169.6 (CO), 169.1 (COOH), 150.5, 150.0, 125.3, 123.7, 117.7, 113.8 (C_arom_), 101.1 (C-1′), 99.4 (C-1), 76.1 (C-4), 72.9 (C-5), 72.4 (C-3), 71.3 (C-2), 70.9 (C-3′), 70.7(C-5′), 69.1(C-2′), 66.6 (C-4′), 61.8 (C-6′), 60.8 (C-6), 56.1 (OMe), 20.8, 20.6, 20.5.

### 3.8. Methyl 3-methoxy-4-(2,3,6,2′,3′,4′,6′-hepta-*O*-acetyl-β-d-lactopyranosyloxy)benzoate (**7**)

Following the general procedure A, compound **7** was obtained as a white powder; yield: 573 mg (66 %): mp 85–90 °C; [α] _20_^D^
_=_ −33.0 (c 0.25, DCM). ^1^H-NMR (300 MHz, CDCl_3_): δ ppm 7.76–7.50 (m, 2H, Ar), 7.10 (d, 1H, *J* = 8.3 Hz, Ar), 5.38–5.37 (m, 1H, H-4′), 5.37–5.28 (m, 1H, H-3), 5.25–5.20 (m, 1H, H-2), 5.17–5.12 (m, 1H, H-2′), 5.05 (d, 1H, *J* = 7.4 Hz, H-1), 4.99–4.96 (m, 1H, H-3′), 4.52–4.51 (m, 2H, H-1′,6a), 4.20–4.08 (m, 3H, H-6b, 6′ab), 3.94–3.88 (m, 2H, H-4, 5′), 3.92 (s, 3H, CO_2_Me), 3.88 (s, 3H, OMe), 3.80–3.75 (m, 1H, H-5), 2.18 (s, 3H), 2.18–1.99 (21H, 7OAc). ^13^C-NMR (75 MHz, CDCl_3_): δ (ppm) 170.3, 170.2, 170.1, 170.0, 169.7, 169.5, 169.1, 166.5, 150.0, 149.8, 126.0, 122.9, 117.9, 113.4, 101.1 (C-1′), 99.6 (C-1), 76.1 (C-4), 72.9 (C-5), 72.4 (C-3), 71.3 (C-2), 70.9 (C-3′), 70.7 (C-5′), 69.0 (C-2′), 66.6 (C-4′), 61.9 (C-6), 60.8 (C-6′), 56.1 (OMe), 52.2 (CO_2_Me), 20.8, 20.6, 20.5. ESI-HRMS: *m*/*z* calcd for C_35_H_44_O_21_, 800.2375; found 818.2675 [M + NH_4_]^+^.

### 3.9. Tert-butyl 3-methoxy-4-(2,3,6,2′,3′,4′,6′-hepta-*O*-acetyl-β-d-lactopyranosyloxy)benzoate (**8**)

Following the general procedure A, compound **8** was obtained as a white powder; yield: 771 mg (58%): mp 98 °C; [α] _20_^D^
_=_ −21.5 (c 0.25, DCM). ^1^H-NMR (300 MHz, CDCl_3_): 7.59–7.51 (m, 2H, H_arom_), 7.19 (d, 1H, *J* = 8.0 Hz,H_arom_),5.38 (d, 1H, *J* = 2.7 Hz, H-4′), 5.31 (dd, 1H, *J* = 9.0Hz, H-3), 5.22 (dd, 1H, *J* = 10.4, *J* =7.8 Hz, H-2), 5.15 (dd, 1H, *J* = 10.4, *J* =7.8 Hz, H-2′), 5.03 (d, 1H, *J* = 7.5 Hz, H-1), 4.99 (dd, 1H,*J* = 10.4, *J* = 3.4 Hz, H-3′), 4.50–4.54 (m, 2H, H-1′,H-6a),4.24–4.05 (m, 3H, H-6b, H-6′ab), 3.88–3.94 (m, 2H, H-4 et H-5′), 3.87 (s, 3H, OMe), 3.81–3.69 (m, 1H, H-5), 2.21–1.94 (21H, 7OAc), 1.60 (s, 9H, 3CH_3_). ^13^C-NMR (75 MHz, CDCl_3_): δ (ppm) 170. 170.2, 170.1, 170.0, 169.7, 169.5, 169.0, 165.2, 149.9, 149.4, 128.0, 122.6, 118.0, 113.4, 101.1 (C-1′), 99.8 (C-1), 81.1 (C-CH_3_), 76.1 (C-4), 72.9 (C-5), 72.5 (C-3), 71.3 (C-2), 70.9 (C-3′), 70.7 (C-5′), 69.0 (C-2′), 66.6 (C-4′), 61.8 (C-6), 60.8 (C-6′), 56.0 (OMe), 28.1 (3CH_3_), 20.7, 20.6, 20.5. ESI-HRMS: *m*/*z* calcd for C_38_H_50_O_21_: 842.2845; found 865.2652 [M + Na^+^].

### 3.10. 3-Methoxy-4-(β-d-lactopyranosyloxy)benzaldehyde (9), (4-Formyl-2-methoxyphenyl β-d-lactopyranoside)

Following the general procedure B, compound **9** was obtained as a white powder; yield: (232 mg, 92 %); mp 215–220 °C. ^1^H-NMR (300 MHz, D_2_O): δ (ppm) 9.73 (s, 1H, CHO), 7.55 (d, 1H, J = 6.7 Hz, Harom), 7.50 (s, 1H, Harom), 7.26 (d, 1H, J = 8.3 Hz, Harom), 5.25 (d,1H, J = 7.5 Hz, H-1), 4.44 (d, 1H, J = 7.6 Hz, H-1′), 3.92–3.96(m, 1H, H-4′), 3.89 (s, 3H, OMe), 3.62–2.84 (m, 9H, H-5′, H-5, H-2, H-3, H-6ab, H-6′ab et H-4), 3.61 (d, 1H, J = 3.1 Hz, H-3′), 3.47 (dd, 1H, J = 9.0 Hz, H-2′). ^13^C-NMR (75 MHz, DMSO-*d*6): δ(ppm) 192.0 (CHO), 152.0, 149.7, 131.0, 125.8, 114.8, 110.8 (Carom), 104.3 (C-1′), 99.2 (C-1), 80.4 (C-4), 76.0 (C-5), 75.4 (C-5′), 75.4 (C-2′), 73.7 (C-3′), 73.2 (C-3), 71.0 (C-2), 68.6 (C-4′), 60.9 (C-6′), 60.4 (C-6), 56.0 (OMe).

### 3.11. Methyl-3-methoxy-4-(3′-*O*-sulfo-β-d-lactopyranosyloxy)benzaldehyde, Sodium Salt (**10**)

Following the general procedure C, compound **10** was obtained as a white powder; yield: (132 mg, quant.): mp 218–232 °C; [α]_20_^D^
_=_ −60.2 (c 0.25, MeOH). ^1^H-NMR (300 MHz, D_2_O): δ (ppm) 9.66 (s, 1H, CHO), 7.47 (d, 2H, *J* = 8.4 Hz, H_arom_), 7.40 (t, 1H, *J* = 5.4 Hz, H_arom_), 7.19 (d, 1H, *J* = 8.7 Hz, H_arom_), 5.19 (d, 1H, *J* = 7.9 Hz, H-1′),4.51 (d, 1H, *J* = 7.9 Hz, H-1′) 4.26 (d, 2H, *J* = 18.4, 8.4, 3.2 Hz, H-3′et H-4′), 3.89–3.98 (m, 2H, H-6), 3.80–3.85 (m, 4H, OMe, H-6′), 3.68–3.78 (m, 5H, H-2′,H-4, H-3 et H-5, H-5′), 3.58–3.68 (m, 1H, H-2). ^13^C-NMR (75 MHz, DMSO-*d_6_*): δ (ppm) 194.7 (CHO), 151.0, 148.8, 130.9, 126.7, 114.7, 111.2 (C_arom_), 102.5 (C-1′), 99.3 (C-1), 79.9 (C-3′), 77.6 (C-4), 75.0, 74.9 et 74.0 (C-3,5,5’), 72.3 (C-2), 66.7 (C-2′), 69.0 (C-4′), 60.9 (C-6′), 59.7 (C-6), 55.7 (OMe). ESI-HRMS: *m*/*z* calcd for C_20_H_28_O_16_S: 556.1093; found 579.0985 [M + Na^+^].

### 3.12. 3-Methoxy-4-(β-d-lactopyranosyloxy)benzylic alcohol (**11**) (4-hydroxymethyl-2-methoxyphenyl β-d-lactopyranoside)

The reduction of compound **9** by NaBH_4_ (1.2 equiv.) in methanol by agitation at room temperature; compound **11** was obtained as a white powder (46 mg, 95%): mp 202 °C; [α]_20_^D^ = −71.5 (c 0.25, MeOH). ^1^H-NMR (300 MHz, D_2_O): δ (ppm) 7.04 (d, 1H, *J* = 8.0 Hz,H_arom_), 6.98 (s, 1H, H_arom_), 6.86 (d, 1H, *J* = 8.2 Hz, H_arom_), 5.03 (d, 1H, *J* = 7.5 Hz, H-1), 4.47 (s, 2H, **CH_2_**OH), 4.37(d, 1H, *J* = 7.6 Hz, H-1′), 3.82–3.87 (m, 1H, H-5 et H-2), 3.77 (s, 3H, OMe), 3.75–3.57 (m, 8H, H-5′, H-4′, H-2′,H-3, H-6_ab_, H-6′_ab_ et), 3.55 (d, 1H*, J* = 2.6 Hz, H-3′), 3.47 (dd, 1H, *J* = 9.0 Hz, H-4). ^13^C-NMR (75 MHz, D_2_O): δ (ppm) 152.0, 148.6, 144.7, 135.8, 120.2, 116.0, 112.0 (C_arom_), 102.9 (C-1′), 100.2 (C-1), 77.9 (C-4), 75.3 (C-5), 74.8 (C-5′), 74.1 (C-3′), 72.4 (C-3 et C-2′), 70.9 (C-2), 68.5 (CH_2_), 63.4 (C-4′), 61.0 (C-6′), 59.7 (C-6), 55.7 (OMe). ESI-HRMS: *m*/*z* calcd for C_20_H_30_O_13_: 478.1686; found 501.1562 [M + Na^+^].

### 3.13. Methyl-3-methoxy-4-(3′-*O*-sulfo-β-d-lactopyranosyloxy)benzylic Alcohol, Sodium Salt (**12**)

The reduction of compound **10** by NaBH_4_ (1.2 equiv.) in methanol gave the compound **12** after 3 h of agitation at room temperature; compound **12** was obtained as a white powder (48 mg, 88%): mp 264 °C. ^1^H-NMR (300 MHz, D_2_O): δ (ppm) 7.09 (d, 2H, *J* = 8.2 Hz, H_arom_), 7.02 (s, 1H, H_arom_), 6.91 (d, 1H, *J* = 8.2 Hz, H_arom_), 5.07(d, 1H, *J* = 7.3 Hz, H-1′),4.51(s, 2H, CH_2-_OH), 4.42 (d, 1H, *J* = 7.9 Hz, H-1′) 4.15 (d, 1H, *J* = 5.6 Hz, H-3′), 3.86–3.92 (m, 2H, H-6ab, et H-6′b), 3.81 (sl, 4H, OMe, H-6′a), 3.49–3.74 (m, 8H, H-2′,H-4, H-4′,H-3 et H-5, H-5′, H-2). ESI-HRMS: *m*/*z* calcd for C_20_H_30_O_16_S: 558.1255; found 557.1196 [M−H]^-^.

### 3.14. Methyl 3-methoxy-4-(β-d-lactopyranosyloxy)benzoate (**13**)

Following the general procedure B, compound **13** was obtained as a white powder; yield: 283 mg (quant.): mp 85–90 °C; *R_f_* 0.26 (CH_2_Cl_2_/MeOH: 75/25). ^1^H-NMR (300 MHz, D_2_O+DMSO-*d_6_*): δ 7.76–7.74 (m, 2H, Ar), 7.31 (d, 1H, *J* = 9.2 Hz, Ar), 5.32 (d, 1H, *J* = 7.7 Hz, H-1), 4.53 (d, 1H, *J* = 7.4 Hz, H-1′), 4.07–3.90 (m, 9H, H-4′,6′,OMe and CO_2_Me), 3.90–3.69 (m, 6H, H-3,4,5,5′,6), 3.66 (m, 2H, H-2,3′), 3.55 (m, 1H, H-2′). ^13^C-NMR (75 MHz, D2O+DMSO-*d_6_*): δ 157.3 (Cq Ar), 150.2 (Cq Ar), 148.5 (Cq Ar), 123.7, 114.8, 112.8, 103.4 (C-1′), 99.3 (C-1), 78.8, 75.6, 75.1, and 74.5 (C-3,4,5,5′), 72.9 (C-3′), 72.7 (C-2), 70.9 (C-2′), 68.5 (C-4′), 61.0 (C-6′), 60.0 (C-6), 56.1 (OMe), 52.7 (CO_2_Me). ESI-HRMS: *m*/*z* calcd for C_21_H_30_O_14_, 506.1636; found 529.1521 [M + Na]^+^.

### 3.15. Methyl 3-methoxy-4-(3′-*O*-sulfo-β-d-lactopyranosyloxy)benzoate, Sodium Salt (**14**)

Following the general procedure C, compound **14** was obtained as a white powder; yield: 45 mg (81%): mp 244–252 °C; [α]_D_-58.8 (c 0.25, MeOH); *R_f_* 0.12 (CH_2_Cl_2_/MeOH: 75/25). ^1^H-NMR (300 MHz, MeOD): δ 7.65 (m, 2H, Ar), 7.26 (d, 1H, Ar), 5.13 (d, 1H, *J* = 7.7 Hz, H-1), 4.56 (d, 1H, *J* = 7.8 Hz, H-1′), 4.37–4.18 (m, 2H, H-3′,4′), 3.98–3.87 (m, 8H, H-6, OMe and CO_2_Me), 3.85–3.77 (m, 2H, H-6′), 3.75 (m, 1H, H-2′), 3.73–3.71 (m, 1H, H-4), 3.70–3.61 (m, 3H, H-3,5,5′), 3.61–3.57 (m, 1H, H-2). ^13^C-NMR (75 MHz, MeOD): δ 156.8 (Cq Ar), 150.6 (Cq Ar), 149.1 (Cq Ar), 123.1, 115.0, 112.7, 103.5 (C-1′), 100.2 (C-1), 80.3 (C-3′), 79.0 (C-4), 75.4, and 75.3 and 74.7 (C-3,5,5’), 73.0 (C-2), 69.5 (C-2′), 67.1 (C-4′), 61.0 (C-6′), 60.1 (C-6), 55.3 (OMe), 51.2 (CO_2_Me). ESI-HRMS: *m*/*z* calcd for C_21_H_30_O_17_S, 586.1204; found 585.1152 [M − H]^-^.

### 3.16. Tert-butyl 3-methoxy-4-(β-d-lactopyranosyloxy)benzoate (**15**)

Following the general procedure B, compound **15** was obtained as a white powder; yield: 122 mg (92 %): mp 206.3 °C; [α]_20_^D^
_=_ −66.0 (c 0.25, MeOH). ^1^H-NMR (300 MHz, CD_3_OD-*d*_4_): δ (ppm) 7.52–7.62 (m, 2H, H_Arom_), 7.19 (d, 1H, *J* = 8.3 Hz,H_Arom_), 5.09(d, 1H, *J* = 7.2 Hz, H-1), 4.42(d, 1H, *J* = 7.4 Hz, H-1′), 3.90 (sl, 4H, H-3′, et OMe), 3.49–3.85 (m, 11H, H-2, 2′, 3, 4, 4′, 5, 5′,6, 6′), 1.60 (s, 9H, 3CH_3_); ^13^C-NMR (75 MHz, CD_3_OD-*d*_4_): δ (ppm) 166.0( CO-C(CH_3_), 150.1, 148.7, 125.8, 123.1, 114.7, 112.5, 103.4 (C-1′), 100.1 (C-1), 81.3 (C-CH_3_), 78.4, 75.6, 75.2, 74.6 (C-3,4,5,5′), 73.2 (C-3′), 72.8 (C-2), 71.1 (C-2′), 68.8 (C-4′), 61.1 (C-6′), 60.0 (C-6), 55.4 (OMe), 27.1 (CH_3_); ESI-HRMS: *m*/*z* calcd for C_24_H_36_O_14_: 548.2105; found 571.1992 [M + Na^+^].

### 3.17. Tert-butyl 3-methoxy-4-(3′-*O*-sulfo-β-d-lactopyranosyloxy)benzoate, Sodium Salt (**16**)

Following the general procedure C, compound **16** was obtained as a white powder; yield: 50 mg (84%): mp 218 °C; [α]_20_^D^ = −104.8 (c 0.25, MeOH). ^1^H-NMR (300 MHz, D_2_O): δ (ppm) 7.34 (d, 1H, *J* = 9.0 Hz, H_arom_), 7.19 (s, 1H, H_arom_), 6.94 (d, 1H,*J* = 8.8 Hz, H_arom_), 5.00 (d, 1H, *J* = 7.6 Hz, H-1′), 4.54 (d, 1H, *J* = 7.7 Hz, H-1′) 4.28 (ddd, 2H,*J* = 22.6, 14.7, 4.7 Hz,H-3′et H-4′), 3.78–4.02 (m, 3H, H-6 et H-5), 3.68–3.78 (m, 7H, OMe, H-6′, H-2′,H-4), 3.54–3.68 (m, 3H, H-2, H-3 et H-5′), 1.44 (s, 9H, 3CH_3_). ^13^C-NMR (75 MHz, D_2_O): δ (ppm) 166.7 (COO(CH_3_)_3_), 149.3, 147.9, 125.8, 123.4, 114.5, 112.5 (C_arom_), 102.6 (C-1′), 99.9 (C-1), 80.1 (C-3′), 78.0 (C-4), 74.1, 74.9, 74.1 (C-3, 5, 5’), 72.41 (C-2), 66.87 (C-2′), 69.13 (C-4′),60.96 (C-6′), 59.96 (C-6), 49.15 (**C**-(CH3)_3_), 27.49 (CH_3_). ESI-HRMS (neg.): *m*/*z* calcd for C_24_H_36_O_17_S: 628.1673; found 627.1611 [M − H]^-^.

### 3.18. 4-E-[(2-Carboxy)ethenyl]-2-methoxyphenyl 2,3,6,2′,3′,4′,6′-hepta-*O*-acetyl-β-d-lactopyranoside (**17**)

A solution of 2 (50 mg, 0.063 mmol), malonic acid (40 mg, 0.381mmol) in mixture of pyridine (0.2 mL) and piperidine (0.01mL) was stirred at 95 °C. After 3 h, the reaction was completed and the mixture was cooled and acidified to PH 1–2 by slow addition of 6M aqueous HCl. The organic phase was extracted twice with dichloromethane, then washed with water to neutralised pH 7 and dried with sodium sulfate. The residue was purified by silica gel column chromatography (MeOH/DCM: 1/9) afforded the corresponding compound **17** as pale-yellow powder, yield: 256 mg (65%): mp 120–125 °C; [α]_20_^D^ = −20.7 (c 0.25, DCM). ^1^H-NMR (300 MHz, CDCl_3_): δ (ppm) 7.68 (d, 1H, *J* = 15.9 Hz, H_a_), 7.06 (s, 3H, 3H_arom_), 6.33 (d, 1H, *J* = 15.9 Hz,H_b_), 5.34 (d,1H, *J* = 3.0 Hz, H-4′), 5.19–5.30(m,2H, H-3 et H-2), 5.08–5.15(m, 1H, H-2′), 4.90–5.01(m, 2H, H-1et H-3′), 4.49–4.52 (m, 2H, H-1′,H-6a), 4.00–4.21 (m, 3H, H-6b, H-6′ab), 3.85–3.91 (m, 2H, H-4 et H-5′), 3.83 (s, 3H, OMe), 3.65–3.77 (m, 1H, H-5), 1.90–2.20 (21H, 7OAc). ^13^C-NMR (75 MHz, CDCl_3_): δ (ppm) 171.7 (COOH), 170.4, 170.3, 170.1, 170.1, 169.8, 169.6, 169.1 (CO), 150.6, 148.1, 130.4, 122.0, 119.0, 111.4 (C_arom_), 146.2 (C-H_a_), 116.5 (C-H_b_), 101.1 (C-1′), 99.8 (C-1), 76.1 (C-4), 72.9 (C-5), 72.5(C-3), 71.3 (C-2), 70.9 (C-3′), 70.7 (C-5′), 69.0 (C-2′), 66.6 (C-4′), 61.8 (C-6′), 60.8 (C-6), 56.0 (OMe), 20.8, 20.6, 20.5. ESI-HRMS: *m*/*z* calcd for C_36_H_44_O_21_: 812.2375; found 835.2267 [M + Na^+^].

### 3.19. 4-E-[(2-Carboxy)ethenyl]-2-methoxyphenyl β-d-lactopyranoside (**18**)

Following the general procedure B, compound **18** was obtained as a white powder; yield: 95 mg (quant.): mp 190–206 °C; [α]_20_^D^ = −201.5 (c 0.25, MeOH); ^1^H-NMR (300 MHz, D_2_O): δ (ppm) 7.22 (d, 1H, *J* = 16.2 Hz, H_a_), 7.18 (s, 1H, H_arom_), 7.06 (dd, 2H, *J* = 16.8, *J* =8.5 Hz, 2H_arom_), 6.34 (d, 1H, *J* = 16.0 Hz,H_b_), 5.03 (d, 1H, *J* = 7.5 Hz, H-1), 4.37 (d, 1H, *J* = 7.5 Hz, H-1′), 3.80–3.88(m, 5H, H-5, H-2, OMe), 3.78–3.53 (m, 9H, H-5′, H-4′, H-3, H-6_ab_, H-6′_ab,_ H-2′et H-3′), 3.46 (dd, 1H, *J* = 9.0 Hz, H-4). ^13^C-NMR (75 MHz, D_2_O): δ (ppm) 175.70 (COOH), 148.6, 146.3, 130.6, 123.2, 121.6, 111.2 (C_arom_), 140.2 (C-H_a_), 115.6 (C-H_b_) 102.9(C-1′), 99.9 (C-1), 77.8 (C-4), 75.3 (C-5), 74.9 (C-3), 74.0 (C-2), 72.4 (C-3′ et C-2′), 70.9 (C-5′), 68.5 (C-4′), 61.0 (C-6′), 59.8 (C-6), 56.8 (OMe), 20.8, 20.6, 20.5. ESI-HRMS: *m*/*z* calcd for C_22_H_30_O_14_: 518.1636; found 541.1522 [M + Na^+^].

### 3.20. 4-(2-Carboxyethyl)-2-methoxyphenyl 2, 3, 6, 2′,3′,4′,6′-hepta-*O*-acetyl-β-d-lactopyranoside (**19**)

To a solution of **17** (200 mg, 0.246 mmol) in THF/MeOH (4/2mL), Pd/C (50 mg) was added under hydrogen atmosphere and stirred for 5 h. The catalyst was removed by filtration through celite and the solvent evaporated under reduced to afford a pale-yellow powder; yield: 200 mg (99%): mp 101 °C; [α]_20_^D^ = −18.1 (c 0.25, DCM); ^1^H-NMR (300 MHz, CDCl_3_): δ (ppm) 6.99 (d, 1H, *J* = 8.1 Hz, H_arom_), 6.74 (d, 1H, *J* = 1.6 Hz,H_arom_), 6.69 (dd, 1H, *J* = 8.2, *J* = 1.7 Hz,H_arom_), 5.35 (d, 1H, *J* = 2.9 Hz, H-4′), 5.27 (t, 1H, H-3), 5.04–5.21 (m, 2H, H-2 et H-2′), 4.97 (dd, 1H, *J* = 10.4, *J* = 3.4 Hz, H-1), 4.89 (d, 1H, *J* = 7.7 Hz, H-3′), 4.47–4.53 (m, 2H, H-1′,H-6a), 4.00–4.21 (m, 3H, H-6b, H-6′ab), 3.83–3.95 (m, 2H, H-4, H-5′), 3.77 (s, 3H, OMe), 3.61–3.72 (m, 1H, H-5), 2.90 (t, 2H, *J* = 7.6 Hz, CH_2a_), 2.64 (t, 2H, *J* = 7.7 Hz, CH_2b_), 1.90–2.20 (21H, 7OAc). ^13^C-NMR (75 MHz, CDCl_3_): δ (ppm) 177.8 (COOH), 170.4, 170.1, 170.0, 169.8, 169.7, 169.1 (CO), 150.5, 144.5, 137.1, 120.2, 120.0, 112.9 (C_arom_), 101.0 (C-1′), 100.5 (C-1), 76.2 (C-4), 72.7 (C-5), 72.6 (C-3), 71.5 (C-2), 70.9 (C-3′), 70.7 (C-5′), 69.1 (C-2′), 66.6 (C-4′), 61.9 (C-6′), 60.8 (C-6), 56.0 (OMe), 30.6 (C-H_a_), 36.0 (C-H_b_), 20.8, 20.6, 20.6, 20.5. ESI-HRMS: *m*/*z* calcd for C_36_H_46_O_21_: 814.2532; found 837.2382 [M + Na^+^].

### 3.21. 4-(2-Carboxyethyl)-2-methoxyphenyl β-d-lactopyranoside (**20**)

Following the general procedure B, compound **19** was obtained as a white powder; yield: 51 mg (94%): mp 190–222 °C; [α]_20_^D^ = −55.8 (c 0.25, methanol); ^1^H-NMR (300 MHz, D_2_O): δ (ppm) 6.96 (d, 1H, *J* = 8.3, H_arom_), 6.86 (s, 1H, H_arom_), 6.72 (d, 1H, *J* =8.3 Hz, H_arom_), 4.94 (d, 1H, *J* = 7.6 Hz, H-1), 4.36 (d, 1H, *J* = 7.6 Hz, H-1′), 3.81 (d, 2H, H-5, H-2), 3.73 (sl, 2H,OMe, H-5′), 3.60–3.72 (m, 5H, H-3, H-6_ab_, H-6′_ab_), 3.60–3.50 (m, 3H, H-4′, H-2′, H-3′), 3.40–3.50 (m, 1H, H-4), 2.73 (t, 1H, *J* = 7.4 Hz, H_a_), 2.40 (t, 1H, *J* = 7.4 Hz,H_b_); ^13^C-NMR (75 MHz, D_2_O): δ (ppm) 175.7 (COOH), 148.6, 146.3, 130.6, 123.2, 121.6, 111.2 (C_arom_), 140.2 (C-H_a_), 115.6 (C-H_b_), 102.9 (C-1′), 99.9 (C-1), 77.8 (C-4), 75.3 (C-5), 74.9 (C-3), 74.0 (C-2), 72.4 (C-3′, C-2′), 70.9 (C-5′), 68.5 (C-4′), 61.0 (C-6′), 59.8 (C-6), 56.8 (OMe), 20.8, 20.6, 20.5. ESI-HRMS: *m*/*z* calcd for C_22_H_32_O_14_: 520.1792; found 543.1674 [M + Na^+^].

### 3.22. Docking Manipulation for Providing Figure 1

DS Biovia Discovery Studio 2020 (https://www.3ds.com) was used for the docking experiments. Galectin-8 N-terminal crystallographic data were obtained from the Protein Data Bank as PDB 3AP6 in which lactose 3′-*O*-sulfate had been co-crystallized. The crystallographic data of p-nitrophenyl lactoside were next obtained from https://pubchem.ncbi.nlm.nih.gov/ as accession no CID 11812612. The two sugar derivatives were superimposed using the tool of the DS software by the tether method and the aglycon (PNP) was next modified into vanillin.

## 4. Conclusions

A series of extended aromatic lactosides harboring vanillin pharmacophores at the anomeric position together with their corresponding O-3′-sulfated analogs were efficiently prepared using PTC and tin acetal-catalyzed stereo-(β-anomer) and regioselective (O-3′-position) transformations, respectively. The *para*-position of the aromatic moiety was further adjusted to afford additional modifications, which would allow evaluating the detailed role of this area upon binding to various galectin family members, especially those (Gal-1, -3, and -8) shown to be particularly sensitive to the presence of sulfation. As previously seen in the case of *E. coli* FimH inhibitors, the design of glycomimetics with optimized aglycons permitted to generate drug-like mannosides of real therapeutic interests [50]. In the present case, the numerous challenges raised for the identification of potent and selective galectin ligands of therapeutic interest are even more exacerbated by the 15 family members having their own physiological function and subtle structural differences. Moreover, the fact that galectins possess extended and shallow binding sites capable of accommodating longer oligosaccharides offers several opportunities in the design of improved ligands. When the actual double modification strategy will be coupled to the third one involving multivalent presentation, further improvement in affinity and selectivity would be achievable, as recently seen when the TD139 clinical phase 2 candidate [10,11,12,13] was coupled to a multivalent protein scaffold [41]. Preliminary data with Gal-3 have been obtained for compounds **13** and **14,** which clearly indicated their potential [15]. Analogous modifications using the *N*-acetyllactosamine scaffold are in preparation and await comparative studies.

## Data Availability

Data sharing not applicable.

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
