# Peer review of "Synthesis of Galectin Inhibitors by Regioselective 3′-*O*-Sulfation of Vanillin Lactosides Obtained under Phase Transfer Catalysis"

_molecules, 2020, doi:10.3390/molecules26010115_

Round 1
Reviewer 1 Report
This concise manuscript describes the synthesis of new lactoside derivatives endowed with potential inhibitory activity against galectins, a family of lectins involved in physiological and pathological processes. The target compounds, obtained by standard chemical transformations and fully characterized, were not submitted to biological assays in the present work (two of them were assayed in a previous work, see ref. 15). This interesting manuscript deserves publications in Molecules after the following minor modifications.
1) The yield of each transformation should be added in all schems.
2) Figure 2: the spectrum of 13 is not clear: the signal at ppm ca. 5.2 does not appear as a doublet as expected for the anomeric H-1 proton. Moreover, this spectrum was recorded for a D2O solution whereas in the Experimental section the solvent for the 1H-NMR spectrum of 13 is D2O-DMSOd6. Unfortunately, copy of the spectrum was not available as Supplementary material.
3) Figure 3: both NMR spectra can be removed since they show the very simple (and easy to prove) reduction of a double bond.
4) References: the number of self-citations is quite high, half of the total number of bibliographic references. Some of them could be removed (some of those dealing with the glycosides synthesis by phase transfer catalysis?).
Author Response
1) The yield of each transformation should be added in all schemes. Done
2) Figure 2: the spectrum of 13 is not clear: the signal at ppm ca. 5.2 does not appear as a doublet as expected for the anomeric H-1 proton. Moreover, this spectrum was recorded for a D2O solution whereas in the Experimental section the solvent for the 1H-NMR spectrum of 13 is D2O-DMSOd6. Unfortunately, copy of the spectrum was not available as Supplementary material. The reviewer’s point is correct, indeed we initially reported the NMR in D2O alone, but since the concentration need for the HSQC and the 13C-NMR need more material for a quick running, we added DMSO. Consequently, we decided to report the data in D2O alone. We placed several more NMR, MS data as Supp material. In addition, he is also correct in commenting that the signal identified as H1 in the Fig. needed corrections since both H1 and H1’ are indeed doublets. The Fig 2 was changed accordingly.
3) Figure 3: both NMR spectra can be removed since they show the very simple (and easy to prove) reduction of a double bond. Done, the spectra are in the SI.
4) References: the number of self-citations is quite high, half of the total number of bibliographic references. Some of them could be removed (some of those dealing with the glycosides synthesis by phase transfer catalysis?). References by the main author (RR) 44-46 were removed. We replaced them by new ref (see below, comment of reviewer 3) to quote the regioselectivity of the tin acetal chemistry.
Reviewer 2 Report
This manuscript describes the synthesis 3’-O-sulfated lactoside derivatives. Regioselective sulfation was achieved by a stannylene acetal method. However, this method is well established and has been applied to very similar aryl lactosides (Ref 47 and ChemBioChem 2003, 4, 640). The sulfanylation reaction of compounds 18 and 20 has not been carried out. Therefore, the corresponding 3’-O-sulfated lactoside derivative are not available. In addition, biological activities of the synthesized compounds have not been evaluated. Considering these points, this manuscript does not include novelty. Thus, it seems unsuitable for the publication in Molecules. Some comments were described below.
- Title is not correct. Sulfation is not “inhibitors”.
- Scheme 1 and 2: Chemical yields must be described in the Scheme.
- Scheme 1: “min” should be added after 2h30.
- The stereoselectivity of the glycosylation reaction should be described. Was only b-isomer obtained?
Author Response
Thus, it seems unsuitable for the publication in Molecules. Some comments were described below. Except for vanillin glucoside, a natural product in foodstuffs, there are no other known vanillin glycoside known, except for the one cited in ref (formerly 48). Hence, the lactoside analogs described herein are novel, even though a brief biological activity has been described by this author in ref 15 for two of them. Given that several other analogs were also in preparation by other group members, we had preferred to keep them all together for biophysical testing (ITC) at the same time, as stated in the original conclusion.
- Title is not correct. Sulfation is not “inhibitors”. The title was corrected to reflect this valuable comment to : “Synthesis of Galectin Inhibitors by Regioselective 3'-O-Sulfation of Vanillin Lactosides Obtained Under Phase Transfer Catalysis”
- Scheme 1 and 2: Chemical yields must be described in the Scheme. Done
- Scheme 1: “min” should be added after 2h30. Done
- The stereoselectivity of the glycosylation reaction should be described. Was only b-isomer obtained? Indeed, this transformation was renowned to be stereo-selective (reviewed in original ref 43, 44. However, a comment was added to this effect: Line 78: “As previously demonstrated [43, 44], these PTC conditions afforded complete stereoselectivity in favor of anomeric inversion from the a-bromide 1 to b-glycosides 5, 7, and 8, ”
Reviewer 3 Report
In their manuscript, Belkhadem et al. report the synthesis of a series of vanilin sulfo-lactosides as potential ligands for the clinically relevant galectin family of glycan-binding proteins.
Their methodology is based on a glycosylation (lactose disaccharides) of vanilin derivatives using phase-transfer catalysis, followed by regioselective sulfation (driven by tin acetal activation).
The targeted family of galectin is highly relevant in biomedical research and good inhibitors remain much needed, as accurately stated by the authors. The reported methodology and the first examples of the vanilin sulfo-lactosides had been previously published elsewhere but the present manuscript expands the scope of this methodology and reports 3 new sulfo lactosides.
I recommend publication after minor revisions:
- line 33: "expressed extra-cellularly". The authors should clarify this sentence - galectins can be secreted but still expressed intra-cellularly.
- line 40: with a number of glycomimetics ? missing word ?
- The choice for the vanilin-based aglycon should be further justified/explained. The low toxicity argument may be insufficient.
- It would be valuable to include a couple of sentences regarding the regioselectivity of the sulfation (explanation/hypothesis).
- Scheme 3 Pyperidine, should be piperidine (same typo in the experimental part
- Fig. 1 Lower panel: legend should describe left/right "sub-panel".
- The docking/model representation is not explained in the experimental section.
- line 151: 2h30
- line 163: 6 hrs / Make consistent with the rest of experimental section
- line 167 missing % sign
- line 211 and elsewhere: HRMS reporting does not follow recommended best practices. For some compounds the HRMS is not reported or reported inappropriately (significant digits).
- General procedure for PTC reaction indicates that the reaction was done in DCM with Na2CO3 (1.3 equiv.) whereas scheme 1 describes the reaction as performed in a mixture of EtOAc / 1M Na2CO3. Please clarify.
- line 352: the described transformation is regioselective but I fail to see the stereoselective aspect of it.
Author Response
- line 33: "expressed extra-cellularly". The authors should clarify this sentence - galectins can be secreted but still expressed intra-cellularly. The sentence was modified as follows: “Given that they are expressed intracellularly together with being present extracellularly by secretion and that…”
- line 40: with a number of glycomimetics ? missing word ? The sentence was modified to: “The field is however only dominated by a few research groups [4-9] amongst which the team of Leffler/Nilsson being clearly dominating with a number of successful glycomimetics, some of which reaching clinical phases (thiodigalactosides, TD139, GB1107) [10-13].
- The choice for the vanilin-based aglycon should be further justified/explained. The low toxicity argument may be insufficient. Sentence added: Line 67 : “Furthermore, even though several hydrophobic aglycons have been advantageous in the binding events using affinity measurements by ITC, we choose the natural vanillin as aglycon because of its well established lack of toxicity, the presence of a large amount of vanillin glucoside (glucovanillin) in foodstuffs [42], and the well-known antioxidant properties of phenolic glycosides.”
- It would be valuable to include a couple of sentences regarding the regioselectivity of the sulfation (explanation/hypothesis). This selectivity is well-known within the carbohydrate community. However, for the sake of non-expert an appropriate sentence was added at line 100 : The regioselectivity of this transformation is well-known and has been explained through the formation of a cyclic stannylene complex at the unique cis-3’,4’-dihydroxyl groups of the galactoside moiety [45-47].
- Scheme 3 Pyperidine, should be piperidine (same typo in the experimental part: done, both parts
- Fig. 1 Lower panel: legend should describe left/right "sub-panel". Done
- The docking/model representation is not explained in the experimental section. This was done by adding the appropriate text on line 358 as follows: Docking manipulation for providing Fig. 1
DS Biovia Discovery Studio 2020 (https://www.3ds.com) was used for the docking experiments. Galectin-8 N-terminal crystallographic data was obtained from the Protein Data Bank as PDB 3AP6 in which lactose 3’-O-sulfate had been co-crystallized. The crystallographic data of p-nitrophenyl lactoside was next obtained from https://pubchem.ncbi.nlm.nih.gov/ as accession no CID 11812612. The two sugar derivatives were superimposed using the tool of the DS software by the tether method and the aglycon (PNP) was next modified into vanillin.
- line 151: 2h30 : done
- line 163: 6 hrs / Make consistent with the rest of experimental section : Done
- line 167 missing % sign : Done
- line 211 and elsewhere: HRMS reporting does not follow recommended best practices. For some compounds the HRMS is not reported or reported inappropriately (significant digits). We completed all missing HRMS data with the usual 4 decimal digits.
- General procedure for PTC reaction indicates that the reaction was done in DCM with Na2CO3 (1.3 equiv.) whereas scheme 1 describes the reaction as performed in a mixture of EtOAc / 1M Na2CO3. Please clarify. This was corrected to solely EtOAc.
- line 352: the described transformation is regioselective but I fail to see the stereoselective aspect of it. The text was amended to Line 358: A series of extended aromatic lactosides harboring vanillin pharmacophores at the anomeric position together with their corresponding O-3’-sulfated analogs were efficiently prepared using PTC and tin acetal-catalyzed stereo- (b-anomer) and regioselective (O-3’-position) transformations, respectively.
Reviewer 4 Report
Ms.Ref.ID: molecules-1005974
Title:”Regioselective 3'-O-Sulfation of Vanilin-based Lactosides as
Inhibitors of Galectins"
Corresponding author: René Roy
SUMMARY: This paper describes the synthesis of a serie of functionalized aromatic lactosides with vanillin pharmacophores at the anomeric position and O-3’-sulfate substituent as potential selective galectin ligands.
RECOMMENDATION: Publish in Molecules after minor revision.
COMMENTS:
The authors perform a straightforward synthesis of new Vanillin-based lactoside derivatives using PTC as a consolidate and efficient methodology for stereoselective glycosidation and the corresponding unprotected lactosides are regioselectively 3’-O-sulfated. These compounds are interesting potential galactin ligands, designed on the basis of the evidence that a few galectin members have been shown to bind preferentially to sulfated glycans.
Therefore, this reviewer recommends publication in Molecules, pending some minor corrections:
- Line 52 Figure 1 Top panel. Lactose structure should be represented as an anomeric mixture of alfa and beta emiacetals.
- Line 86 Scheme 1. The reaction conditions should be changed in agreement with what is reported in the general procedure (line 147-150).
- Line 377 References. In most the references (with the exception of Ref. 3, 4, 26, 27) DOI codes are missing.
Author Response
Therefore, this reviewer recommends publication in Molecules, pending some minor corrections:
- Line 52 Figure 1 Top panel. Lactose structure should be represented as an anomeric mixture of alfa and beta emiacetals: Done
- Line 86 Scheme 1. The reaction conditions should be changed in agreement with what is reported in the general procedure (line 147-150): Done
- Line 377 References. In most the references (with the exception of Ref. 3, 4, 26, 27) DOI codes are missing. We left this part to be completed by the editorial staff, as usual.
Round 2
Reviewer 2 Report
The authors revised the manuscript according to the reviewers' suggestions and made improvements. However, for some questions, the authors ignored the comments. While the purpose of this study should be the synthesis of sulfated vanillin lactosides and their galectin inhibitory activity, but derivatives 18 and 20 are not sulfated. Moreover, the inhibitory activity of the synthesized compound has not been evaluated at all. This report is partial and should be reviewed again after adding the required items.